Gliding toward an understanding of the origin of flight in bats

http://orcid.org/0000-0002-2401-2044 Burtner Abigail E. 1
M. Grossnickle David 1 2
Santana Sharlene E. 1
Law Chris J. 1 3 chrislaw@utexas.edu
1 University of Washington , Seattle, WA , United States
2 Oregon Institute of Technology , Klamath Falls, OR , United States
3 University of Texas at Austin , Austin , United States
Pyenson Nicholas
Electronic publication date: 2024 Jul 25
Publication date: 2024
Volume: 12
Electronic Location ID: e17824
Received 2024 Feb 28; Accepted 2024 Jul 8
Copyright: © 2024 Burtner et al.
Copyright year: 2024
Copyright holder: Burtner et al.
License: This is an open access article distributed under the terms of the Creative Commons Attribution License, which permits unrestricted use, distribution, reproduction and adaptation in any medium and for any purpose provided that it is properly attributed. For attribution, the original author(s), title, publication source (PeerJ) and either DOI or URL of the article must be cited.
License URL: https://creativecommons.org/licenses/by/4.0/

Keywords: Adaptive landscape, Ecomorphology, Macroevolution, Ornstein–Uhlenbeck modeling, Phylogenetic comparative methods

Funding: Mary Gates Endowment and the American Society of Mammalogists Grant-in-Aid of Research Award National Science Foundation IOS-2017738 National Science Foundation DBI-1906248 and DBI–2128146 Gerstner Family Foundation and the Richard Gilder Graduate School at the American Museum of Natural History University of Texas Early Career Provost Fellowship Abigail E. Burtner was supported by the Mary Gates Endowment and the American Society of Mammalogists Grant-in-Aid of Research Award; David M. Grossnickle and Sharlene E. Santana were supported by National Science Foundation IOS-2017738; and Chris J. Law was supported by National Science Foundation DBI-1906248 and DBI–2128146, the Gerstner Family Foundation and the Richard Gilder Graduate School at the American Museum of Natural History, and a University of Texas Early Career Provost Fellowship. The funders had no role in study design, data collection and analysis, decision to publish, or preparation of the manuscript.

==============================
Bats are the only mammals capable of powered flight and have correspondingly specialized body plans, particularly in their limb morphology. The origin of bat flight is still not fully understood due to an uninformative fossil record but, from the perspective of a functional transition, it is widely hypothesized that bats evolved from gliding ancestors. Here, we test predictions of the gliding-to-flying hypothesis of the origin of bat flight by using phylogenetic comparative methods to model the evolution of forelimb and hindlimb traits on a dataset spanning four extinct bats and 231 extant mammals with diverse locomotor modes. Our results reveal that gliders exhibit adaptive trait optima (1) toward relatively elongate forelimbs that are intermediate between those of bats and non-gliding arborealists, and (2) toward relatively narrower but not longer hindlimbs that are intermediate between those of non-gliders and bats. We propose an adaptive landscape based on limb length and width optimal trends derived from our modeling analyses. Our results support a hypothetical evolutionary pathway wherein glider-like postcranial morphology precedes a bat-like morphology adapted to powered-flight, setting a foundation for future developmental, biomechanical, and evolutionary research to test this idea.

Introduction

Powered flight has evolved at least four times over the course of organismal evolution, consistently serving as a gateway to ecomorphological and species diversification (Heard & Hauser, 1995; Hunter, 1998; Jetz et al., 2012; Nicholson, Ross & Mayhew, 2014). As the only mammals capable of powered flight, bats have correspondingly specialized body plans that include highly derived limbs. These specialized morphologies are understood to be the result of adaptations for the demands of flight; the skeletal elements of bat forelimbs are elongated to support flight membranes and increase aerodynamic efficiency, and bat hindlimbs are not only specialized for hanging and catching prey in flight (Norberg, 1990; Gunnell & Simmons, 2005; Giannini, 2012), but also contribute to flight itself (Cheney et al., 2014).

The evolutionary origin of flight in bats remains a subject of ongoing research and debate due to an uninformative fossil record linking bats to their non-flying ancestors. Specifically, there are no known transitional species in the fossil record to offer evidence of potential morphological precursors to flight in modern bats (Brown et al., 2019). Nevertheless, bat flight is widely hypothesized to be the result of an ancestral “trees-down” transition from non-gliding tree-dweller (hereafter, ‘arborealist’) to glider to flyer (Norberg, 1985b; Bishop, 2008; Simmons et al., 2008), albeit this idea has not been tested using phylogenetic comparative methods. It should be noted that all phylogenomic evidence shows that bats are not closely related to any extant glider (Upham, Esselstyn & Jetz, 2019; Álvarez-Carretero et al., 2022); bats are nested within Laurasiatheria where neighboring clades (i.e., carnivorans, ungulates, and pangolins) are predominantly adapted to terrestrial locomotion. Laurasiatheria is distantly related to placental gliders, including dermopterans and rodents, located in the Euarchontoglires superorder.

Following the arborealist-to-glider-to-flier hypothesis, the ancestors of bats–which originated about 70–60 million years ago (Teeling et al., 2000; Upham, Esselstyn & Jetz, 2019; Álvarez-Carretero et al., 2022)–are hypothesized to have evolved more elongate limbs, followed by patagia (wing membranes) on the armwing to facilitate gliding, and finally membranes in the handwing to enable powered flight. The armwing consists of stylopodal (i.e., humerus) and zeugopodal (i.e., ulna and radius) bones, while the handwing consists of autopodal bones (i.e., the metacarpals and phalanges). The evolution of the handwing is thought to be the key innovation that catalyzed bats’ adaptive radiation (Vaughan, 1970; Sears et al., 2006; Amador, Simmons & Giannini, 2019), resulting in the evolution of over 1,500 extant species (Simmons & Cirranello, 2022). Along with the forelimbs, the hindlimbs structure the plagiopatagium, which provides the center of lift for both bats and all gliding mammals (Swartz, Bishop & Aguirre, 2005; Jackson & Schouten, 2012). The uropatagium, which extends behind the hind legs and can encompass the tail in bats and some species of gliding mammals, also affects airfoil shape and thus lift. Furthermore, the hindlimbs are involved in quadrupedal locomotion in some species of bats (Schutt & Simmons, 2005) and their role in hindlimb suspension has been hypothesized to be another key adaptation in facilitating the transition to powered flight (Giannini, 2012).

Previous work has examined many biological features associated with the evolution of bat flight, such as sensory adaptations (e.g., echolocation or vision), aerodynamics (e.g., aerofoil and flight performance), ecology (e.g., diet, habitat, roosting), and development (e.g., patagia development and digit elongation) (Bishop, 2008; Cooper & Tabin, 2008; Giannini, 2012; Anthwal et al., 2023; Feigin et al., 2023). Morphological approaches to understanding the origin of bat flight further include analyses of skeletal dimensions in the context of development (Adams, 1992, 2008; Sears et al., 2006; Nojiri et al., 2023), biomechanics (Swartz, Bennett & Carrier, 1992; Swartz, 1997; Swartz & Middleton, 2008), and functional morphology (Vaughan, 1970). These studies often focus on aspects of the bat forelimb (e.g., Amador, Simmons & Giannini, 2019) or hindlimb (e.g., Riskin, Bertram & Hermanson, 2005), but seldom both in tandem in a phylogenetic context (but see Swartz & Middleton (2008) and Nojiri et al. (2023)). Here, we test predictions based on the gliding-to-flying hypothesis of the origin of bat flight by modeling the evolution of forelimb and hindlimb traits on a dataset spanning four extinct bats and 231 extant mammals with diverse locomotor modes including ground-dwellers, arborealists, gliders, and flyers.

Following the arborealist-to-glider-to-bat hypothesis, we test two predictions regarding the evolution of limb morphologies using an Ornstein-Uhlenbeck evolutionary model-fitting approach. First, we predict that selection will lead the forelimb skeletal traits of arborealists, gliders, and bats towards three separate adaptive trait optima/attractors (i.e., θ parameters of fitted OU models, which are commonly interpreted as the locations of adaptive peaks in morphospace; hereafter “optima”), where the glider forelimb optimum will be intermediate in value to those of arborealists and bats. If supported, this demonstrates that glider morphology may fall on an intermediate adaptive peak between bat and arborealist morphologies. Specifically, we predict wider and longer skeletal element optima values progressing from arborealists to gliders to bats. Relatively longer bones (especially in the handwing) and larger-diameter bones (especially in combination with thinner, highly mineralized walls), may give bats enough resistance to torsion during powered flight (Swartz, Bennett & Carrier, 1992) while still increasing their patagial area. Glider optima with intermediate forelimb length and width optima between arborealist and bat optima may represent an intermediate adaptive zone between the anatomical adaptations required for tree-dwelling (Cartmill, 1985) and those required for powered flight (Norberg, 1990). Support for this prediction would suggest selective pressures for flight influenced the evolution of bat forelimb skeletal elements from glider-like forelimbs.

Second, we predict that bat hindlimb optima will exhibit a distinct trait optimum (i.e., adaptive peak) separate from ground-dwelling, arboreal, and gliding mammals. Previous work has considered bat hindlimbs to be reduced in size (e.g., Farnum, Tinsley & Hermanson, 2008; Simmons et al., 2008; Louzada, Nogueira & Pessôa, 2019), whereas gliders exhibit relatively lengthened hindlimbs (Grossnickle et al., 2020; Linden et al., 2023; Rickman et al., 2023). Therefore, the relative lengths and widths of the hindlimb bones are not well characterized between these aerial groups as well as in relation to arborealists. Here, we aim to identify where bat hindlimb adaptive optima lie relative to other mammals. Specifically, we predict narrower and longer skeletal element optima values progressing from arborealists to bats and gliders. We predict overlapping hindlimb optima between gliders and bats because relatively longer and lower-diameter hindlimb bones in both groups relative to arborealists and ground-dwellers may decrease weight while still generating lift. Support for this prediction would suggest that shared selective pressures for less-quadrupedal motion (i.e., as gliding and flying both incorporate hindlimbs into the wing membrane) influenced the evolution of bat hindlimb skeletal elements from ancestors with glider-like hindlimbs. Overall, this study offers the first phylogenetically informed test of morphological data to investigate whether the adaptive optima of arborealists, gliders, and flyers illustrate a possible route across adaptive peaks in the evolution of mammalian flight.

Materials and Methods

Portions of this text were previously published as part of a preprint (Burtner et al., 2024).

Morphological data

We compiled linear measurements of 14 forelimb and 15 hindlimb traits (Fig. 1) from 231 extant mammals (1–2 specimens per species) across 21 orders (78% of mammalian orders) and 73 families (47% of mammalian families). Of these, 27 representative species of bats (48 specimens) were measured for the flyer group. These spanned both Yangochiroptera and Yinpterochiroptera suborders, five families, and fifteen genera ranging in mass from 5 to 700 g, thus broadly covering both the phylogenetic and physical diversity of Chiroptera. Linear measurements were obtained from specimens held at the Burke Museum of Natural History and Culture or from previously published work (Chen & Wilson, 2015; Grossnickle et al., 2020; Weaver & Grossnickle, 2020; Pevsner, Grossnickle & Luo, 2022). All specimens measured for this study were fully mature, determined by the closure of exoccipital–basioccipital and basisphenoid–basioccipital sutures on the cranium and ossification of all vertebrae and limb bones. Specimens were a mixture of male, female, and unknown sexes. We size-corrected all linear measurements by calculating log-shape ratios of each trait measurement (Mosimann, 1970) by log10-transforming the mean trait value divided by the cube root of body mass prior to running all statistical analyses. Unfortunately, most specimens lacked accompanying body mass data; therefore, we compiled species-level body mass data from the PanTHERIA database (Jones et al., 2009) and the Handbook of the Mammals of the World book series (Wilson & Mittermeier, 2009).

Figure 1 Morphological measurements and phylogeny.

We conducted analyses based on the mammalian phylogeny from Upham, Esselstyn & Jetz (2019) across 14 forelimb linear measurements and 15 hindlimb measurements for each specimen. The forelimb measurement consisted of scapula height and length (sh, sl), humerus length (hl), distal width (hdw), proximal width (hpw), and mid-shaft width (hsw), radius length (rl), ulna length and olecranon length (ul, uol), and third manual digit metacarpal length and width (mcl, mcw), proximal phalanx length and width (ppl, ppw), and intermediate phalanx length (ipl). The hindlimb measurements consisted of pelvis, ilium, and ischium length (pel, il, isl); femur length (fl), distal width (fdw), and midshaft width (fsw); tibia length (tl), proximal length (tpw), and midshaft width (tmw); fibula length (fbl), and third pedal digit metatarsal length and width (mtl, mtw), proximal phalanx length and width (ppxl, ppxw), and intermediate phalanx length (ipxl). We measured only the third digit in this dataset (and not, for example, the fifth even though that might be of biological interest) to build on previous datasets in the literature to generate a large enough dataset for evolutionary modeling. The skeletal figures are modified from Chen & Wilson (2015).

We also collected the 29 measurements for four extinct bats (Onychonycteris finneyi, Icaronycteris index, Hassianycteris messelensis, and Tachypteron franzeni) using ImageJ (Schneider, Rasband & Eliceiri, 2012) on published full-skeleton images that include scale bars (Storch, Sigé & Habersetzer, 2002; Gunnell & Simmons, 2005; Simmons et al., 2008). We were only able to include these four extinct bats because we could only obtain a complete set of measurements from these species. O. finneyi, I. index, and H. messelensis have been recovered as stem bats in phylogenetic reconstructions and thus provide critical context for the evolution of powered flight in the absence of transitional bats in the fossil record (O’Leary et al., 2013; Rietbergen et al., 2023). Meanwhile, T. franzeni is generally thought to be a crown bat, so it is also interesting to include as the only non-stem fossil bat in our analyses (Storch, Sigé & Habersetzer, 2002; Ravel et al., 2016; Rietbergen et al., 2023). We estimated the body masses of extinct bats using a linear regression between body mass and the geometric mean of all linear measurements from the extant bats of our dataset, and used those body mass estimations to size-correct our fossil data again using log-shape ratios. We included fossil bats in our morphological dataset because, although extinct bats have a high degree of morphological similarity to extant bats (Norberg, 1985b; Bishop, 2008; Simmons et al., 2008), the fossil bat data may alter model estimations of the adaptive zone between flyers and gliders in an informative way. For example, the inclusion of fossils could make the flyer-glider trait optima more similar or more disparate by adding more phylogenetic history information and thus inform the understanding of the adaptive landscape underlying the evolution of mammalian flight.

To visualize the distribution of locomotor groups in the forelimb and hindlimb morphospaces, we performed principal component analyses (PCA) of the morphometric data using the princomp function in the R package stats (R Core Team, 2021).

Locomotor data

We categorized species into four locomotor groups: ground dwellers (n = 117), arborealists (n = 71), gliders (n = 16), and flyers (n = 27) following previous work (Chen & Wilson, 2015; Grossnickle et al., 2020; Weaver & Grossnickle, 2020; Pevsner, Grossnickle & Luo, 2022). The ground-dwelling mammals included diverse ambulatory, cursorial, saltatorial, and fossorial taxa. This ‘catch-all’ category was intentional as we are interested in investigating the arboreal to gliding to flying specialization trends only. Our arborealist category was also broad in that it includes both fully arboreal (tree-dwelling) and scansorial taxa. Despite these broad categories, ground-dwellers and arborealists show less variability in their trait optima for both the forelimb and hindlimb compared to gliders and flyers (see Results). Therefore, we do not view the ground-dwelling or arboreal optima as being pulled away from other groups in a forced way. Gliders included tree-dwelling mammals with derived morphologies (i.e., patagia and elongated limbs) for gliding locomotion (Grossnickle et al., 2020). Flyers included bats only.

Phylogenetic comparative analyses

We used evolutionary models (Hansen, 1997; Butler & King, 2004) to examine the influence of locomotor modes on the evolution of forelimb and hindlimb morphologies in our extant-only and extant+extinct datasets. We first tested the hypothesis that locomotor modes influenced the evolution of the forelimb and hindlimb using multivariate evolutionary models on our extant datasets (Butler & King, 2004; Hansen, 1997) in the R package mvMORPH (Clavel, Escarguel & Merceron, 2015). To reduce the dimensions of our data, we conducted a principal component analysis (PCA) on our 14-trait forelimb dataset and 15-trait hindlimb dataset. We retained the first three principal components (PCs) of our forelimb dataset (91.8% of the forelimb variation) and hindlimb dataset (85.3% of hindlimb variation). We then fit six multivariate models to the scores of these three PCs. We fit a multivariate Brownian motion model (mvBM), which models trait variance in a random walk over time; a single-regime multivariate Ornstein-Uhlenbeck model (mvOU1), in which scores of each PC evolve towards a single trait optimum (θ parameter of OU models), giving a single mean optimum for all locomotor groups; and multivariate OU models (mvOUM) with a given number of optima corresponding to selective regimes (based on locomotor modes). To test our hypotheses on overall limb evolution, we fit a four-peak OU model (mvOUMloc4) for distinct optima among ground-dweller, arborealist, glider, and flyer regimes. To test alternative hypotheses, we fit three three-peak models (mvOUMloc3a-c) with distinct combinations of the four locomotor groups. The mvOUMloc3a model tests for distinct optima among ground-dweller, arborealist, and a combined glider-flyer regimes. The mvOUMloc3b model tests for distinct optima among ground-dweller, combined arborealist-glider, and flyer regimes. The mvOUMloc3c model tests for distinct optima among a combined ground-dweller-arborealist, glider, and flyer regimes.

To account for uncertainty in phylogenetic topology, branch lengths, and ancestral character states (Fig. 1), we used the make.simmaps function in the R package phytools (Revell, 2012) to simulate 10 stochastic character maps across 1,000 trees randomly drawn from the posterior distribution of “completed” trees from Upham, Esselstyn & Jetz (2019), resulting in 10,000 character maps of locomotory regimes. We randomly sampled 250 trees and fit all six models to these trees. We assessed relative model support using small sample corrected Akaike weights (AICcW). Models with ΔAICc scores less than two were considered supported and interpreted as evidence of locomotor groups influencing species’ adaptive zones. We acknowledge that using a subset of PCs instead of the full dataset could bias our results (Adams & Collyer, 2018; Uyeda, Caetano & Pennell, 2015). However, our results remained consistent when we increased the number of PCs in our model-fitting analyses to the first four or five PCs. Further, we used simulations to assess whether we had adequate power to accurately distinguish between complex mvOU models from Brownian motion (Boettiger, Coop & Ralph, 2012). We also calculated the phylogenetic half-lives (ln(2)/ α) of the best fitting model to assess the responsiveness of locomotor regimes to adaptive peaks (Hansen, 1997).

If the mvOUMloc4 model is best-supported, we interpreted this as each locomotor regime having its own distinct adaptive optimum. This result alone does not provide evidence for the gliding origin of bats. However, if gliders are located at an optimum between arborealists and flyers (i.e., if arborealist < glider < flyer optima or arborealist > glider > flyer optima), there is support for an evolutionary “pathway” between the two locomotor modes relative to an ancestral arborealist or ground-dwelling state (i.e., one could theoretically bridge adaptive peaks via these pathways). Therefore, if the optima gradually change in value (as opposed to random changes in value) from arborealists to gliders to bats, our hypothesis that bats evolved from gliding ancestors is supported.

If the mvOUMloc3a model is best-supported, we interpret this as gliders+flyers having a shared adaptive optimum and arborealists and ground-dwellers having their own distinct evolutionary optima. This model provides the clearest support for flyers evolving from gliding ancestors because lineages of both groups are evolving towards the same shared adaptive optima. Therefore, if this model best fit the data, our hypothesis is supported.

If the mvOUMloc3b model is best-supported, we interpret this as arborealists+gliders having a shared adaptive optimum and flyers and ground-dwellers having their own distinct evolutionary optima. There is phylogenetic justification for this grouping because all gliders evolved from ancestrally arboreal groups (e.g., gliding squirrels evolved from tree squirrels; see tree in Fig. 1). These groups are similar to the 4-group model in that if arborealists+gliders have an optimum between those of ground-dwellers and flyers, there is support for an evolutionary “pathway” between the two locomotor modes relative to an ancestral ground-dwelling state.

Finally, if mvOUMloc3c model is best-supported, we interpret this as ground-dwellers+arborealists having a shared adaptive optimum and flyers and gliders having their own distinct evolutionary optima. As with the mvOUMloc4 and mvOUMloc3b models, if gliders have an optimum between those of ground-dwellers+arborealists and flyers, there is support for an adaptive “pathway” between the two locomotor modes relative to an ancestral arborealist or ground-dwelling state, providing evidence for our hypothesis that flyers evolved from gliding ancestors.

We also examined the influence of locomotor modes on the evolution of each individual forelimb and hindlimb trait in our extant-only and extant+extinct datasets using univariate evolutionary models. For each of the 29 traits, we fit a Brownian motion model (BM), a single-optimum Ornstein-Uhlenbeck model (OU1), and four multiple-peak OU (OUM) models. The OUM models included OUMloc4, OUMloc3a, OUMloc3b, and OUMloc3c, featuring the same locomotor regimes as the multivariate mvMORPH models. All models were fit using the R package OUwie (Beaulieu et al., 2012). To maintain consistency with the mvMORPH analyses, we applied a single selection parameter ( α) and rate parameter ( σ2) across all selective regimes while allowing optima (θ) to vary between regimes (Hansen, 1997; Butler & King, 2004). For traits in which an OUM model was the best-supported model, we computed 95% confidence intervals using parametric bootstrapping of the model parameters with the OUwie.boot function in the R package OUwie (Beaulieu et al., 2012). We randomly sampled 100 of our character maps and performed 10 bootstrap replicates per stochastic character map sampled.

We repeated the above multivariate and univariate model fitting approaches on a merged dataset that includes extant species and the four fossil species. We first generated a MCC tree from the 1,000 trees randomly drawn from the posterior distribution of “completed” trees from Upham, Esselstyn & Jetz (2019) and manually placed the four extinct bats on the phylogeny as a polytomy with extant bats at 69.8 MYA, which was the average branch age of bats based on the posterior distribution of “completed” trees. Although Tachypteron franzeni is often hypothesized to fall within crown bats (Storch, Sigé & Habersetzer, 2002; Ravel et al., 2016; Rietbergen et al., 2023), the exact relationship with extant bats remains unresolved (e.g., Jones, Beard & Simmons, 2024). Therefore, we conservatively include T. franzeni in our polytomy instead of within extant bats. Regardless, the position of T. franzeni is unlikely to change the model results because all extinct taxa show fairly derived traits (compared to gliders, arborealists, and generalist species included) (e.g., T. franzeni lies consistently with both extant and extinct bats in morphospace and Fig. S1). We then simulated 50 stochastic character maps on the phylogeny and fit all four models across the 50 stochastically mapped trees to account for uncertainty in the ancestral character states.

Lastly, additional processes outside the scope of our investigation among arboreal, gliding, and flying transitions may also influence limb evolution, resulting in evolutionary shifts in limb morphology that are not associated with the evolution of gliding or flying. These hidden processes are not captured by our hypothesis-testing framework described above. Therefore, we also used PhylogeneticEM to determine if additional shifts of limb evolution occur along branches in a pattern not predicted by our locomotor categorical scheme. The PhylogeneticEM approach does not use a priori regime assignments (i.e., locomotor mode assignments are not incorporated into analyses). We used a scalar OU model that infers the full evolutionary rate matrix and accounts for correlations within multivariate datasets (i.e., PC1–PC3 for both forelimb and hindlimb datasets) using the R package PhylogeneticEM version 1.4.0 (Bastide et al. 2018).

Results

Forelimb morphospace and evolutionary model-fitting

PC1 explains 81.5% of the forelimb variation and separates flyers (both extant and extinct) from the remaining three locomotor modes (Fig. 2A; Table S1A). PC1 also slightly separates gliders, particularly dermopterans, from arborealists and ground-dwellers. This axis describes the relative lengths of the armwing (humerus, radius, and ulna) and the handwing (metacarpals and phalanges). PC1 varies across relatively shorter (−PC1) to elongate (+PC1) bones, revealing a trend within the dataset in the elongation of multiple forelimb bone elements from arborealists to gliders to flyers. PC2 explains 7.8% of the morphological variation and primarily describes the relative widths of the long bones and hands. This variation of wide to narrow bones mostly occurs within locomotor groups, especially within the diverse ground-dweller group. PC3 explains 2.9% of the morphological variation with +PC3 dominated by metacarpal length, ulna olecranon length, and scapula height and −PC3 dominated by intermediate phalanx length and humerus distal width. Fossil bats overlap with extant bats in morphospace, which is consistent with the lack of transitional forms in the bat fossil record.

Figure 2 Forelimb and hind limb PCAs.

Morphospaces consist of PC1 and PC2 of 14 forelimb traits (left panels) and 15 hind limb traits (right panels). The extant+extinct morphospace is shown for the (A) forelimbs and (B) hind limbs. The extant-only morphospace of (C) the mvOUMloc3b optima (large icons) overlaid on the forelimb morphospace and the (D) the mvOUMloc4 optima (large icons) overlaid on the hind limb morphospace. Cartoons of a (E) quadrupedal mammal skeleton (Rattus norvegicus) versus a (F) bat skeleton (Pipistrellus abramus) are shown where dark gray shaded bones on the cartoon skeletons correspond to those measured to generate the morphospace.

The mvOUMloc3b model, which assigns arborealists and gliders to the same regime, was the best-fitting model for the extant multivariate forelimb dataset (PCs 1–3) (AICcW = 0.99, Table 1A) with an average phylogenetic half-life of 15.2 million years (myr) among PCs 1–3. Distinct optima were supported for these three locomotor groups, with PC1 and PC2 optima scores increasing in value from ground-dweller to arborealist+glider to flyer hindlimbs (Fig. 2C). Repeating the multivariate model-fitting with the addition of the four extinct flyer species yielded a mvOUMloc4 model as the best fit (mAICcW = 0.99; Table S6A) with an average phylogenetic half-life of 127.2 myr among PCs 1–3. However, the optima exhibited unrealistic magnitudes (Table S6B).

Table 1 AICcW scores from multivariate model-fitting analyses.

Bolded rows indicate the best fitting model.

(A) Forelimb	AIC	ΔAICc	mAICcW	(B) Hindlimb	AIC	ΔAICc	mAICcW	
mvBM1	−94.42	225.50	0.00	mvBM1	−134.64	230.05	0.00	
mvOU1	−248.73	71.20	0.00	mvOU1	−330.72	33.97	0.00	
mvOUMloc3a	−306.82	13.10	0.00	mvOUMloc3a	−358.41	6.28	0.04	
mvOUM loc3b	−319.93	0.00	0.99	mvOUMloc3b	−353.51	11.17	0.00	
mvOUMloc3c	−308.77	11.16	0.00	mvOUMloc3c	−358.10	6.59	0.03	
mvOUMloc4	−302.35	17.57	0.00	mvOUM loc4	−364.69	0.00	0.92	
Note:

Comparisons of the fits of multivariate models to the first three principal components of the (A) forelimb dataset and (B) hindlimb dataset. Bolded rows indicate the best fitting model.

When we examined each forelimb trait individually using univariate models, we found that the OUMloc4 model was the best-fitting model for six of the 14 forelimb traits (AICcW = 0.51–1.00), OUMloc3b was the best-fitting model for five traits (AICcW = 0.36–0.65), OUMloc3c was the best fitting model for metacarpal length (AICcW = 0.63), and OUMloc3a was the best-fitting model for ulna olecranon length (AICcW = 0.58; Table S2A). The OU1 model was the best fit for the proximal phalanx width of the third manual digit (AICcW = 0.51). Armwing (i.e., humerus and radius) length traits have optima (from the best-fitting OUMloc4 models) that gradually increase from ground-dwellers to arborealists to gliders to flyers; parametric bootstrapping revealed little to no overlap in the 95% confidence intervals of optima between most locomotor groups (Figs. 3B and 3C). Optima of armwing widths (from the best-fitting OUMloc3b models) indicated that flyers exhibit relatively wider humeri at the distal end, proximal end, and mid-shaft compared to all other locomotor groups (Figs. 3G–3I; Table S3A). Flyers exhibit relatively higher optima for handwing length traits (i.e., longer third manual metacarpal, proximal phalanx, and intermediate phalanx) than all other locomotor groups (Figs. 3D–3F; Table S3A). For each, glider optima were intermediate between arborealist and flyer optima. For handwing width traits, flyers exhibit relatively higher OUMloc3b optima for metacarpal widths compared to all other locomotor groups (Fig. 3J), but no differences in proximal phalanx width as evidenced by OU1 support for this trait. An adaptive optimum for a flyer-glider combined group in the OUMloc3a models was supported for ulna olecranon length only (Table S2A). In total, however, glider optima (in combination with arborealists) were between the 95% confidence intervals of ground-dweller and flyer optima for all traits except scapula length/width, humerus proximal width, and metacarpal width (Figs. 3A, 3G, 3J; Table S3A). For these last trait optima, glider+arborealist optima were statistically the same as that of ground-dwellers while flyers exhibited higher optima than all other locomotor groups (Figs. 3A, 3G, 3J; Table S3A).

Figure 3 (A–J) Forelimb trait optima from model-fitting of univariate traits.

Forelimb OUM optima (θ) as a size-corrected trait value (log-shape ratio) with 95% confidence intervals (Tables S3a) for the best-fitting models (ΔAIC = 0) for 10 traits. The AICcW of the model is on top of each panel (Table S2a).

Adding fossil flyers into forelimb model-fitting analyses yielded overall slightly more support for OUMloc3a models compared to extant-only modeling; scapula height and humerus proximal width demonstrate support for a flyer-glider combined group (Table S4A). OUMloc4 and OUMloc3b models were still widely supported as the best-fitting model for six traits and five traits, respectively; OU1 was still the best-fitting model for proximal phalanx width (Table S4A). Overall, the length and width optima values for the forelimb were very similar between extant+extinct and extant only-modeling (Table S5A).

Hindlimb morphospace and evolutionary model-fitting

PC1 explains 47.5% of hindlimb variation and more positive PC1 scores describe the relative elongation (+PC1) of the fibula, metatarsal, and pedal proximal and intermediate phalanges (Fig. 2B; Table S1B). PC1 slightly separates gliders and flyers, with gliders falling towards +PC1 and flyers falling towards −PC1, indicating glider hindlimb autopodia are overall more elongated than flyers’. Arborealists and ground-dwellers are spread across PC1. PC2 explains 17.8% of the variation, where +PC2 is associated with relatively lengthened pedal proximal phalanges, intermediate phalanges, and ischium and −PC2 with relatively narrower pedal proximal and intermediate phalanges. PC2 loosely separated ground-dwellers, arborealists, and the glider-flyer group. PC3 explains 11.3% of the morphological variation with +PC3 dominated by ilium length, femur length, and fibula length and −PC3 dominated by metatarsal width, intermediate phalanx width, and proximal phalanx width. These loadings show that variation between the four locomotor groups mainly occurs throughout the hindlimb autopodia region. It should be noted that many bats have incomplete fibulae, so perhaps the reason that fibula length had a higher contribution than tibia length to the differences across PCs 1–3 was due to this anatomical difference in bats; regardless, since fibula and tibia length loadings were closely correlated, it appears that in this PCA the zeugopodal length of mammals also contributes substantially to overall morphological differences out of the traits measured (Table S1).

The mvOUMloc4 model was the best-fitting model for the extant multivariate hindlimb dataset (PCs 1–3) (AICcW = 0.92, Table 1B) with an average phylogenetic half-life of 0.05 myr among PCs 1–3. Distinct optima were supported for the four locomotor regimes, with PC1 and PC2 optima scores decreasing in value from ground-dweller to arborealist to glider to flyer hindlimbs, albeit with overlap (Fig. 2D). Adding in the four extinct flyer species yielded again the mvOUMloc4 model as the best fit, and very little change in the optima values (Table S6) with an average phylogenetic half-life of 15.2 myr among PCs 1–3.

When we examined each hindlimb trait individually using univariate models, we found that the OUMloc4 model was the best-fitting model for three of 15 traits, all of which were hindlimb long bone lengths (AICcW = 0.33–0.60; Table S2B). Specifically, gliders have relatively longer tibia and femoral optima compared to flyers and the other locomotor groups (Figs. 4B and 4C, Table S3B ). Fibula length was the third hindlimb trait supported by OUMloc4, but fibulae are rudimentary in most bats. The OUMloc3a model best fit three traits (AICcW = 0.36–0.52), all of which were widths: femur shaft width, metatarsal width, and pedal proximal phalanx width (Table S3B). Parametric bootstrapping of these OUMloc3a-supported models revealed that the glider-flyer group had relatively narrower femur shafts, metatarsals, and pedal proximal phalanges optima than the other locomotor groups (Figs. 4F, 4I, 4J; Table S3B). The OUMloc3b model was the best-fitting model for four of the 15 traits (AICcW = 0.44–0.71): ilium, ischium, metatarsal, and intermediate proximal phalanx length (Table S2B). Flyers had significantly smaller relative ilium and ischium length optima than other locomotor groups. Gliders+arborealists had metatarsal and intermediate proximal phalanx optima intermediate between flyers and ground-dwellers (Figs. 4D and 4E; Table S3B). The OUMloc3c model best fit three traits (AICcW = 0.42–0.48): pelvis length, femur distal width, and tibia proximal width (Table S2B). In the latter two traits, gliders exhibited intermediate width optima between flyers (which were smaller) and ground-dwellers+arborealists (which were larger). For pelvis length, like ilium and ischium length, there was no clear trend in the direction of the optima values across locomotor groups (Table S3B).

Figure 4 (A–J) Hind limb trait optima from model-fitting of univariate traits.

Hind limb OUM optima (θ) as a size-corrected trait value (log-shape ratio) with 95% confidence intervals (Tables S3b) for the best-fitting models (ΔAIC = 0) for 10 traits. The AICcW of the model is on top of each panel (Table S2b).

Adding fossil flyers into hindlimb model-fitting analyses yielded only one difference compared to extant-only modeling: pelvis length was now best fit with a OUMloc3b instead of OUMloc3c model (Table S4B). The length and width optima for the hindlimb were very similar between extant+extinct and extant-only modeling (Table S5B).

Evolutionary shifts in limb morphology

The PhylogeneticEM model indicated seven evolutionary shifts in the forelimb and 12 shifts in the hindlimb (Fig. 5). These shifts occur along the branches of named clades. In the forelimb, evolutionary shifts occurred along monotremes, three clades of “moles” (i.e., Notorycitidae, Chrysochloridae, and Talpidae), Chiroptera, Antilopinae, and Primatomorpha. In the hindlimb, evolutionary shifts occurred along Cingulata, Tenrecidae, Chiroptera, Primatomorpha, Hystricognathi rodents, Dipodidae, and marsupials. Further shifts in hindlimb evolution occur within marsupials including Notoryctidae, Peramelidae, Myrmecobiidae, Potoroidae, and Macropodidae.

Figure 5 (A and B) Clade-specific evolutionary shifts in limb morphology across the mammalian phylogeny identified by PhylogeneticEM.

Shifts are represented as colored circles, and branches on the phylogenies are colored according to each regime.

Discussion

Forelimb trait evolution

We found evidence that mammalian glider forelimb morphologies are evolving toward adaptive peaks that are intermediate between those of arborealists and flyers. Our multivariate forelimb results and univariate length trait results provide support for the hypothesis that selective pressures for flight influenced the evolution of flyer forelimbs from glider-like forelimbs. Using multivariate evolutionary models, we found a three optima model with distinct selective regimes for ground-dwellers, arborealists+gliders, and flyers as the best fit to the first three PCs of our data (Table 1A). The arborealist+glider optima are intermediate between those of flyers and ground-dwellers (Fig. 2C). PC1 is mostly explained by the variation in the forelimb bone lengths and PC2 by their widths, creating a coarse picture that forelimb bones likely become relatively longer and possibly wider from ground-dwellers to arborealists+gliders to flyers. Results from our univariate evolutionary models are consistent with these findings, revealing that glider forelimb optima are intermediate between flyer and arborealist optima for length traits (Figs. 3A–3F), but only partially for width traits (Figs. 3G–3J) across both the armwing and handwing (See Table 2 for a breakdown of whether each univariate model supports our hypothesis). The phylogenetic half-lives of the best multivariate model (15.2 myr) and mean of best univariate models of each trait (18.4 myr) are short relative to Chiroptera (ca. 70 MYA), providing support that the forelimb traits are strongly pulled toward distinct locomotor peaks across the adaptive landscape. Furthermore, variation in phylogenetic half-lives from univariate models (Table 2) suggest that these traits are pulled towards distinct peaks either over different adaptive slopes or at different rates.

Table 2 Summary of the best-fitting univariate OUwie models.

(A) Forelimb	
Forelimb trait	Extant	θ interpretation	α	λ	Supports hypothesis?	Extant+extinct	α	λ	
Scapula length	3b	ground~arb+glide<bat	0.128	5.40	No	3b	0.134	5.16	
Scapula height	4	ground~arb~glide<bat	0.074	9.38	No	3a	0.064	10.79	
Humerus length	4	ground<arb<glide<bat	0.048	14.37	Yes	4	0.050	13.86	
Radius length	4	ground<arb<glide<bat	0.043	16.08	Yes	4	0.075	9.26	
Ulna length	4	ground<arb<glide<bat	0.070	9.96	Yes	4	0.101	6.84	
Ulna olecranon length	3a	ground>arb>glide+bat	0.084	8.30	Yes	4	0.102	6.81	
Metacarpal length	3c	ground+arb<glide<bat	0.010	69.78	Yes	3b	0.013	51.70	
Prox. Phalanx length	4	ground<arb<glide<bat	0.022	30.86	Yes	4	0.024	29.40	
Intermed. Phalax length	4	ground<arb<glide<bat	0.032	21.52	Yes	4	0.036	19.03	
Humerus proximal width	3b	ground~arb+glide<bat	0.096	7.24	No	3a	0.094	7.35	
Humerus shaft width	3b	ground<arb+glide<bat	0.097	7.17	Yes	3b	0.087	7.94	
Humerus distal width	3b	ground<arb+glide<bat	0.035	19.60	Yes	3b	0.028	24.80	
Metacarpal width	3b	ground>arb+glide<bat	0.034	20.12	No	3b	0.031	22.47	
Prox. Phalanx width	OU1	--	--	--	No	OU1	--	--	
(B) Hindlimb	
Hindlimb trait	Extant	θ interpretation	α	λ	Supports hypothesis?	Extant+extinct	α	λ	
Pelvis length	3c	ground+arb~glide>bat	0.051	13.56	No	3b	0.049	14.19	
Ilium length	3b	ground~arb+glide>bat	0.043	15.95	No	3b	0.042	16.66	
Ischium length	3b	ground~arb+glide>bat	0.047	14.60	No	3b	0.049	14.09	
Femur length	4	ground<arb<glide>bat	0.035	19.74	No	4	0.029	23.79	
Tibia length	4	ground<arb<glide>bat	0.024	28.58	No	4	0.023	29.94	
Fibula length	4	ground~arb<glide>bat	0.031	22.24	No	4	0.024	28.86	
Metatarsal length	3b	ground>arb+glide>bat	0.017	41.24	Yes	3b	0.016	42.10	
Pedal prox. Phalanx length	OU1	--	--	--	No	OU1	--	--	
Intermed. Pedal phalanx length	3b	ground<arb+glide<bat	0.107	6.46	Yes	3b	0.108	6.44	
Femur distal width	3c	ground+arb>glide>bat	0.076	9.08	Yes	3c	0.079	8.75	
Femur shaft width	3a	ground~arb>glide+bat	0.070	9.96	Yes	3a	0.061	11.36	
Tibia midshaft width	OU1	--	--	--	No	OU1	--	--	
Tibia proximal width	3c	ground+arb>glide>bat	0.113	6.15	Yes	3c	0.115	6.01	
Metatarsal width	3a	ground>arb>glide+bat	0.123	5.64	Yes	3a	0.135	5.13	
Pedal prox. Phalanx width	3a	ground>arb>glide+bat	0.115	6.05	Yes	3a	0.119	5.81	
Note:

Summary of the best-fitting univariate OUwie models with interpretation of the optima (θ) in the context of our initial predictions (see Methods) and alpha (α, strength of selection) and phylogenetic half-life (λ, time required to move halfway from the ancestral state to the optimum) values for the (A) forelimb and (B) hindlimb.

Flyers exhibit trait optima indicating evolution of relatively longer forelimb bones compared to all other locomotor groups (Figs. 2A, 3A–3F) particularly in the handwing, which exhibited rapid evolution as a necessary adaptation for bats (Vaughan, 1970; Sears et al., 2006; Amador, Simmons & Giannini, 2019). Increased bone lengths may enhance the surface area of the patagium for greater lift (Norberg, 1985b). Flyers also displayed evidence of evolution toward relatively wider bones in the armwing compared to all other locomotor groups (Figs. 3G–3J) consistent with findings that bat humeri and radii exhibit significantly larger diameters compared to nonvolant mammals (Swartz & Middleton, 2008). These relatively wider proximal forelimb bones combined with thinner yet highly mineralized walls provide resistance to bending and torsion during wing flapping (Swartz, Bennett & Carrier, 1992; Papadimitriou, Swartz & Kunz, 1996). Furthermore, the transition from relatively more robust proximal ends of the humerus to relatively less robust distal ends may also reflect greater muscle mass closer to the center of rotation of the wing joints, which would greatly decrease flight inertial power and facilitate economical flight (Norberg, 1985a; Swartz, 1997). In the handwing, we found flyers exhibit relatively thicker third metacarpals (Fig. 3J) that corresponded with its relatively increased length (Fig. 3D) compared to nonvolant mammals. In contrast, the proximal phalanx length of the third manual digit was relatively longer compared to nonvolant mammals whereas its width was best supported with an OU1 model, consistent with previous work showing that the proximal phalanx diameter is not significantly smaller in bats compared to nonvolant mammals (Swartz & Middleton, 2008). As a result, bats exhibit relatively thick bone cortices to combat greater bending moments associated with this relatively increased elongation (Swartz, Bennett & Carrier, 1992; Papadimitriou, Swartz & Kunz, 1996; Swartz, 1997). Increased cortical thickness along with the reduction of mineralization of the handwing work together to resist bending failure while simultaneously maintaining flexibility during flying (Papadimitriou, Swartz & Kunz, 1996).

In gliders, we found evidence that forelimb bone length optima are largely intermediate between ground-dwellers and flyers. Gliders tend to exhibit relatively greater length optima in most forelimb bones compared to ground-dwelling and arboreal mammals (Thorington & Heaney, 1981; Thorington & Santana, 2007; Grossnickle et al., 2020; Linden et al., 2023; Rickman et al., 2023) but do not reach the lengths of flyer forelimb bones. Nevertheless, like flyers, the longer bones expand the surface area of the patagium (Thorington & Heaney, 1981; Thorington & Santana, 2007) and enable further adjustment of their patagium during flight, leading to more efficient aerodynamics, increased agility, and maneuverability (Thorington, Darrow & Anderson, 1998; Zhao et al., 2019). Forelimb bone widths in gliders, while clearly intermediate between those of ground-dwellers and flyers for some traits (Figs. 3H, 3I) share an optimum with arborealists (shown by the OUMloc3b support for these traits; Figs. 3G–3J) and overlap with ground-dwellers for other trait optima (Figs. 3G, 3J). Together, these results suggest gliders may maintain high relative widths while continuing to elongate forelimb bones, supporting previous findings that the evolutionary elongation of glider forelimbs is not also accompanied by slimming of the bones (Grossnickle et al., 2020).

Hindlimb trait evolution

We found mixed evidence that mammal glider hindlimb morphologies are intermediate to those of arborealists and flyers. Using our multivariate modeling approach, we found distinct optima for ground-dwellers, arborealists, gliders, and flyers that best fit the first three PCs of our hindlimb data (Table 1B). The trait loadings indicate that the autopodial bones dictate the distribution of PC scores in PCs 1 and 2 (Table S1B). However, there is much overlap among the optima of ground-dwellers, arborealists, and gliders in morphospace (Fig. 2D).

The univariate models uncovered further patterns: 1) width optima of hindlimb bones were consistently narrower in flyers and gliders compared to arboreal mammals (Figs. 4F–4J), and 2) there was no pattern from arboreal to glider to flyer in the hindlimb length optima. Univariate modeling revealed that the widths of hindlimb bones tend not to differ between flyers and gliders (Figs. 4F–4J) because the widths of hindlimb bones in flyers and gliders exhibited either a single optimum (i.e., OUMloc3a) separate from ground-dwelling and arboreal mammals (Figs. 4F, 4I, 4J) or an overlap in confidence intervals of optima between flyers and gliders (Figs. 4G, 4H). This shared flyer-glider adaptive zone suggests that flyers and gliders exhibit narrower hindlimb optima, especially in the autopod region (Figs. 4F–4J). In contrast, univariate modeling revealed distinct optima in long bone lengths of the hindlimb among the four locomotor regimes, in which gliders exhibited relatively longer femoral and tibial optimal lengths compared to flyers, arborealists, and ground-dwellers (Figs. 4B and 4C). Unlike bat forelimbs, bat long bones in the hindlimb (i.e., the femur and tibia) are not the longest compared to the other locomotor regimes; in fact, the confidence intervals of optimal lengths largely overlap between flyers and arborealists (Table S3B). Furthermore, flyers exhibit relatively shorter metatarsals but relatively longer pedal intermediate phalanges compared to all other locomotor regimes, whereas these autopodial lengths do not differ between gliders and arborealists (Figs. 4D and 4E). The different evolutionary patterns of hindlimb bones demonstrate the unique locomotor diversity among the four regimes. Although gliders and flyers are both aerial, these behaviors require different functional morphologies. In gliders, relatively longer hindlimbs are needed to increase the surface area of the patagium in tandem with relatively longer forelimbs to generate lift during gliding (Thorington & Heaney, 1981). In contrast, in bats, the relative lengthening of the forelimb supports most of the wing membranes whereas the hindlimb, specifically the ankle, serves as an anchor to the wing membrane. Bat hindlimbs are still used for flight by helping modulate lift, drag, or pitch (Cheney et al., 2014).

Overall, gliders appear to be evolving relatively lengthened hindlimb bones while bats are evolving more gracile hindlimb bones. Considering these results, we reject our prediction that flyer and glider hindlimbs overlap in their adaptive optima. This finding may point to the fact that modern gliders are completely distinct evolutionarily and potentially even morphologically from the gliding ancestors of bats. These results may highlight the pitfalls of modeling modern gliders as proxies for ancestral ones instead of invalidating our hypothesis that bats evolved from ancestral gliders. Further support of these hypotheses is provided by the relatively short phylogenetic half-lives of the best multivariate model (0.05 myr) and average of the best univariate models of each trait (15.3 myr) compared to the age of Chiroptera (ca. 70 MYA).

Incorporating fossil bats

The morphospace positions of fossil bats in the forelimb PCA confirm extinct bats’ morphological similarity to extant bats in the degree of forelimb specialization (Fig. 2A). Amador, Simmons & Giannini (2019) modeled aerodynamic traits instead of skeletal traits and found that the Eocene bat Onchonycteris finneyi occupies an area of aerofoil morphospace between extant gliders and bats. In our morphospace, O. finneyi is the closest fossil bat to gliders across PC1 (Fig. S1A), congruent with Amador, Simmons & Giannini’s (2019) findings. Tachypteron franzeni is the only extinct species included that is hypothesized to be part of an extant bat family (Emballonuridae or Miniopteridae), but its forelimb morphology lies outside the range of living bats along PC2 (Fig. S1A) (Storch, Sigé & Habersetzer, 2002). A caveat is that no species of either crown Emballonuridae or Miniopteridae were included in this study.

Extinct flyers occupy the same region of hindlimb morphospace as extant flyers, but three of four species overlap clearly with the region of morphospace of non-flyers, a higher proportion compared to that of extant bats (Fig. 2B). In this hindlimb morphospace, T. franzeni is the extinct species that most clearly overlaps with extant flyers (Fig. S1B). Based on its intermediate limb traits, Simmons et al. (2008) suggested that O. finneyi was likely more proficient at climbing than extant bats and may have employed quadrupedal locomotion and under-branch hanging. Interestingly, O. finneyi and Desmodus rotundus (the common vampire bat) are almost indistinguishable in the hindlimb morphospace (Fig. S1B). Vampire bats are known for their adept quadrupedal locomotion and climbing (e.g., Riskin, Bertram & Hermanson, 2005), so O. finneyi’s similar morphology may lend support to more walking-capable early bats. However, the linear measurements we used here may not reflect the complex adaptations required for these behaviors (e.g., ninety-degree joint rotations for under-branch hanging, Riskin, Bertram & Hermanson, 2005), so the placement of these species in morphospace leaves many possibilities open for ancestral locomotor behaviors.

Including extinct flyers in our model-fitting analyses as the closest morphologies we have to flyer-glider intermediaries did not significantly alter estimations of the adaptive zone between flyers and gliders; the inclusion of extinct species appeared to make the flyer-glider trait optima marginally more similar for the forelimbs and had no change for the hindlimbs (Table 2). The addition of extinct species to the multivariate model-fitting analysis shifted support from mvOUMloc3b to mvOUMloc4, suggesting that the glider optimum is potentially distancing itself from arborealists’ and towards flyers’ (Table S6A). Unfortunately, the extant+extinct mvOUMloc4 model predicted extremely negative optima values, which represent unrealistic morphologies that are potentially artifactual, making it difficult to gain more insight into this result (Table S6B). The addition of extinct species to univariate forelimb model-fitting analyses yielded overall slightly more support for OUMloc3a models compared to extant-only modeling (Table S4A). The addition of extinct species to hindlimb model-fitting analyses yielded overall no change in support for the mvOUMloc4 model or OUMloc3a models compared to extant-only modeling (Tables S6, S4B). Regardless, the length and width trends are the same as above between locomotor groups across the fore- and hindlimb (Table S5).

Limb trait optima and the adaptive landscape

Our results provide insight on the potential adaptive landscape connecting arboreal, gliding, and flying locomotion (Fig. 6). We found that glider forelimb optima fall intermediate between those of arborealists and flyers in their lengths but only partially in their widths. Except for long bone lengths, glider hindlimb optima are intermediate between ground-dweller/arborealist and flyer optima or even overlap with flyer optima. Together, these results emphasize that selection on key length/width traits may be pulling glider traits towards a flying adaptive zone yet that certain glider traits (e.g., femur and tibia lengths) are derived specifically for gliding morphology. Bishop (2008) proposed that the high degree of convergence among gliders suggests functional constraints on the evolution of powered flight from gliding; however, more recent work found that gliding lineages display incomplete convergence (species retain unique morphologies even while evolving some similarities; Leal, Knox & Losos, 2002; Stayton, 2006). For gliders, there is little evidence that gliding is under strong selective pressures toward a single-optimum gliding morphology (Grossnickle et al., 2020). The clustering of bat species in forelimb morphospace (Fig. 2A) may represent a narrow adaptive peak within which forelimb morphologies are highly conserved. In contrast, the broad glider morphospace may represent a gradually inclined selection plane, with gliders still far from a peak (Grossnickle et al., 2020). This broad adaptive slope could represent the foothill of a steep flyer peak such that gliders are very slowly evolving towards a flyer peak (Fig. 6A). The evolutionarily oldest extant glider lineages, dermopterans (colugos) and anomalurids (scaly-tailed squirrels), have evolved farthest from ancestral arborealists (Grossnickle et al., 2020) although some extinct gliders may have experienced relatively greater change from ancestral arborealists (e.g., Meng et al., 2017; Luo et al., 2017). Here, the dermopteran species Cynocephalus volans and Galeopterus variegatus are the gliders closest to the flyer forelimb region of morphospace (Figs. 2A; S1A), suggesting a trajectory in morphospace from gliding to flying. Further, colugos are known as the “mitten gliders” because of their interdigital membranes, which, along with their common carpal morphology with bats (Simmons & Geisler, 1998), reflects hypotheses that interdigital webbing was key to evolving powered flight (Anderson & Ruxton, 2020; but see Gardner & Dececchi, 2022 for an alternate view).

Figure 6 Hypothetical adaptive landscapes of forelimb and hind limb skeletal relative length and width evolution.

Hypothetical fitness peaks are shown for arborealists, gliders, and bats for a broad summary of our conclusions from the skeletal trait optima results discussed. The “length” and “width” in the figure refer to the lengths and widths of individual limb elements. For the (A) forelimb, we depict arborealists on a relatively steep adaptive peak at relatively lower length/width optima, gliders on a broad, shallow peak/incline at relatively higher length but not width optima, and bats on a very steep peak at relatively higher length and width optima. For the (B) hind limb, we depict arborealists again on a fairly steep adaptive peak at relatively higher width but not length optima, bats on a shallow steep peak at relatively higher length and lower width optima, and gliders on a peak similar to and overlapping with bats’ at relatively higher length but similar width optima.

Our results also suggest that both flyers and gliders exhibit broad adaptive optima (i.e., having large confidence intervals) in the hindlimb due to gliders sharing an optimum or overlapping optima with flyers, or being intermediate between non-flyers and flyers for many traits (Figs. 2D, 4; Amador, Simmons & Giannini, 2019). These broad adaptive optima could represent overlapping peaks for gliders and flyers separated only by a shallow fitness valley driven by the gliders’ elongated femur and tibia lengths (Fig. 6B). The hypothetical adaptive landscape is therefore valuable because the transition from quadrupedalism to hindlimb suspension has been hypothesized to enable forelimbs to evolve more freely since they would no longer be weight-bearing structures (Bishop, 2008). Indeed, there is both quadrupedal suspension and hindlimb suspension in dermopterans, resulting in a tradeoff in the ability to walk quadrupedally upright (Fujiwara, Endo & Hutchinson, 2011; Granatosky, 2018). In the PC1 and PC2 hindlimb morphospace, G. variegatus is the only glider that overlaps with the flyer morphologies (Figs. 2B; S1B), reflecting a hindlimb adaptive optimum potentially more easily attained than that of flyers’ specialized forelimbs (Fig. 6).

Our modeling approach offers a first step towards support for the hypothesis that selection may more strongly pull certain limb traits towards a flyer adaptive zone. In particular, gliders’ intermediate forelimb length and overlapping hindlimb width optima between that of arborealists and flyers could be the traits driving lineages across fitness valleys due to their greater effect in, for example, the functionality of the gliding/flying apparatus. Short phylogenetic half-lives relative to the crown age of Mammalia provide additional evidence of distinct locomotor peaks. However, to complete the tentative adaptive landscape presented here (Fig. 6), these results could be strengthened by more comprehensive coverage of the lengths/widths across more mammal species and biomechanical studies tying these traits to specific functions and selective pressures.

Importantly, we acknowledge that the evolutionary transitions from arboreal to gliding to flying are not the only processes that dictate the adaptive landscape of mammalian limb evolution. Additional processes not explicitly tested here may also influence the evolution of the forelimb and hindlimb. Using PhylogeneticEM, we found that evolutionary shifts in limb morphology occur along the branches of named clades (Fig. 5). In the forelimb, we find evidence that these shifts are loosely structured based on locomotor mode. Antelopes (Antilopinae), colugos and primates (Primatomorpha), and bats (Chiroptera) exhibited evolutionary shifts in the forelimb, and all exhibit adaptations to cursorial, arboreal, and flying locomotor modes, respectively (Polly, 2007). Furthermore, three of the shifts occur on branches along three clades (marsupial moles, Notoryctidae; golden moles, Chrysochloridae; and moles, Talpidae) that independently evolved subterranean fossoriality. Their adaptations for stout, robust forelimbs are well documented (Polly, 2007). Evolutionary shifts in the hindlimb can also be partially explained by distinct locomotor modes including arboreal, flying, and saltatorial adaptations in colugos and primates (Primatomorpha), bats (Chiroptera), and jerboas (Dipodidae). Five shifts in hindlimb morphology also occur within marsupials, a result that may be unsurprising considering that marsupials display greater hindlimb disparity compared to placentals (Pevsner, Grossnickle & Luo, 2022). Overall, these results highlight that confounding effects from ecological adaptations and phylogenetic structure can result in clade-specific shifts across the adaptive landscape of mammals and other vertebrates (Uyeda & Harmon, 2014; Felice et al., 2019; Law, 2021; Law et al., 2022; Law, Hlusko & Tseng, 2024).

Conclusions

Our findings complement earlier work investigating the evolution of bat powered flight by testing the selective regimes acting on bone dimensions across the mammalian forelimb and hindlimb. Our results reveal that the relatively lengthened forelimbs in gliders are intermediate to those of bats and arborealists, whereas bats and gliders exhibit similarly relatively narrower hindlimbs compared to arborealists. Thus, solely based on morphometrics, the traditional highly specialized view of glider/bat forelimbs and the reduced view of bat hindlimbs may represent more malleable adaptive zones than previously thought. There is a need for future studies to test the biomechanical consequences of such trait optima, alongside more synergistic studies taking into account bat and gliders’ complex physiology and development to test these hypotheses regarding the evolution of mammalian powered flight.

Supplemental Information

Supplemental Information 1 Raw forelimb data.

Supplemental Information 2 Raw hindlimb data.

Supplemental Information 3 Supplementary Data.

Supplemental Information 4 R script for mvMorph models for forelimb.

Supplemental Information 5 R script for mvMorph models for forelimb with extinct bats.

Supplemental Information 6 R script for mvMorph models for hindlimb.

Supplemental Information 7 R script for mvMorph models for hindlimb with extinct bats.

Supplemental Information 8 R script for OUwie models for univariate limb traits.

HDW is used as an example

Supplemental Information 9 R script to conduct bootstrapping on OUwie model parameters.

HDW is used as an example

Supplemental Information 10 R script to obtain parameters from OUwie models.

HDW is used as an example

Supplemental Information 11 R script to make locomotor SIMMAPs (hindlimb dataset).

Supplemental Information 12 R script to make locomotor SIMMAPs (forelimb dataset).

Supplemental Information 13 R script to perform PhylogeneticEM.

Supplemental Information 14 Rdata of forelimb data.

Supplemental Information 15 Rdata of forelimb data with extinct bats.

Supplemental Information 16 Rdata of hindlimb data.

Supplemental Information 17 Rdata of hindlimb data with extinct bats.

We thank Jeff Bradley (UWBM) for extensive access to specimens, members of the “Tempo & Mode in Isolation” discussion group for assistance with data collection, and Lucas N. Weaver for giving permission for our use of his original skeletal illustrations of Rattus norvegicus. We thank Nicholas Pyenson, Matthew Jones, and one anonymous reviewer for helpful suggestions.

Additional Information and Declarations

Competing Interests

Author Contributions

Data Availability

The authors declare that they have no competing interests.

Abigail E. Burtner conceived and designed the experiments, performed the experiments, analyzed the data, prepared figures and/or tables, authored or reviewed drafts of the article, and approved the final draft.

David M. Grossnickle conceived and designed the experiments, authored or reviewed drafts of the article, and approved the final draft.

Sharlene E. Santana conceived and designed the experiments, authored or reviewed drafts of the article, and approved the final draft.

Chris J. Law conceived and designed the experiments, performed the experiments, analyzed the data, prepared figures and/or tables, authored or reviewed drafts of the article, and approved the final draft.

The following information was supplied regarding data availability:

The raw data are available in the Supplemental Files.

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
