# Peer review of "Gliding toward an understanding of the origin of flight in bats"

_PeerJ, doi:10.7717/peerj.17824_

## Round 0.1 · original submission · Major Revisions

This manuscript has now received two thorough and very helpful reviews. There is good overlap in the recommended changes from both reviewers. While the scope of their recommended revisions differed, they largely agree on substantive changes that will require some work, but no major re-analyses should the authors be able to explain the identified challenges: i.e., the justification for the taxonomic scope of their dataset; include multiple phylogenetic hypotheses for bat origins or explain how different hypotheses would affect their analyses; and reporting model parameters for OU analyses. The reviewers also helpfully modest line edits that should be easily addressed. Overall, I think these revisions amount to moderate reviews, and so the editorial decision should be interpreted as "major leaning toward minor revisions. A revised manuscript will probably be sent to one or both of these original reviewers.

·

Basic reporting

Good. Consider moving table S7 to main body of manuscript, otherwise no comment.

Experimental design

No comment

Validity of the findings

No comment

Additional comments

This is a wonderful new analysis and advancement in the understanding of the origins of bat flight, and addresses an important gap in our knowledge of the transition from a (presumably) arboreal ancestor to fully flighted bats. The model-fitting approach the authors used seems to be an appropriate way to compare trends in the evolution of bat limbs with those of gliders, arborealists, and ground-dwelling mammals, and the sample size used in their analyses was impressive. I applaud the authors for including four fossil taxa in their analysis (especially given the notably poor bat fossil record), although I would like to have seen them elaborate on the choice and relevance of these taxa for their analyses (see below). I recommend only minor revisions, comments below:

Main comments
• Please provide a brief justification for the selection of extant and fossil bat taxa used in the analysis. Were the extant taxa chosen to broadly sample living bat diversity? More importantly, provide evolutionary context for the fossil bats used in this study. I think the inclusion of fossil bats into this analysis is an interesting—and valuable—addition to this manuscript. The authors should expand a bit on the inclusion of the four specific taxa they chose and their hypothesized relationships to living bats. Onychonycteris finneyi, Icaronycteris index, and Hassianycteris messelensis are commonly considered to be outside of the bat crown group and are almost always recovered as such in analyses of bat relationships. Their position as stem bats and inclusion in this analysis is arguably essential for our understanding of the origin of bat flight in lieu of true “transitional” bats. Tachypteron franzeni is generally thought to be a crown bat and, although it has been less frequently included in phylogenetic analyses, this position has been consistently recovered in recent analyses. The authors should expand on these points and the relevance of these particular taxa for their study in the paragraph beginning on line 124, or in a new paragraph in the Methods section.
• Lines 231-233: I don’t necessarily disagree with this approach, but why the choice to place all four fossil taxa in a polytomy, and why the choice of 69.8 mya? Tachypteron, at least, has pretty consistently been suggested/recovered to belong among the crown group (e.g., Storch et al. 2002; Ravel et al. 2016; Rietbergen et al. 2023). Does placing it in a polytomy along with the other fossil taxa + crown bats have any bearing on the results? The authors should elaborate on these choices and whether or not they would impact their results.
• Figure S7 is so important for the take home message. This should go in the main body and not supplemental info.

Minor comments
Lines 189-190: This should probably be changed to “This result alone does not provide evidence for the gliding origin of bats.”
Line 239: perhaps specify dermopterans in particular among gliders
Lines 248-249: In Fig. 2a, two fossil bats are outside of the range of living bats along PC2. One is Onychonycteris finneyi (according to Fig. S1), is the other Icaronycteris index? This might be worth calling attention to if so, as those are typically thought to be the most postcranially “primitive” fossil genera. (Perhaps also reference this in the Discussion around line 435).
Lines 270-271: typo? Fig. 3b-f makes it appear that glider optima are intermediate between arborealist and flyer optima.
Line 291: No mention of tibia length in this section? How does the incomplete fibula of most bats impact these results?
Line 314: I would change to read “most bats”. Among this dataset, I think at least Desmodus, Tadarida, Onychonycteris, and Icaronycteris all have complete fibulae.
Line 326: “where” should be “which”
Line 384: “ground-dwelling optima mammals” is odd phrasing, consider revising.
Line 488: Galeopterus is misspelled (also in supplemental material)
Line 489: should this be Fig. 2a?
Supplemental Figure S1: Galeopterus and Onychonycteris misspelled in figure and figure caption
Supplemental Figure S1: Last line of figure caption should be changed to refer to Onychonycteris finneyi as “most postcranially primitive” fossil bat or something similar (many fragmentary bat dentitions and a few skeletons are older)
Supplemental Table S7: This table should probably be moved to main body. Also not clear what shaded boxes in table represent

Reviewer 2 ·

Basic reporting

I thought this manuscript had clearly stated hypotheses, expectations for those hypotheses, and thorough analyses. The raw data files were shared, though no code was provided.

Experimental design

I understand that a sufficiently large sample size is necessary for modeling, but I am a bit confused about some of the sampling and breakdown of locomotor groups. First, the sentence “This ‘catch-all’ category was intentional as we are interested in investigating the arboreal to gliding to flying specialization trends only.” does not make sense to me. Something like a bear, or hyena, or ungulate feel quite different compared to the other mammals in your sample (and are quite different from each other as well), such that they may naturally pull any “ground-dwelling” optima away from other groupings in a more forced way. The inclusion of many large bodied mammals in your sample is also confusing, given the arborealists, gliders, and flyers you’re focusing on are all small-bodied. To avoid any potential allometric effects, it may be better to stick to mammals of similar size to the groups in question. Second, “Our arborealist category was also broad in that it includes both fully arboreal (tree-dwelling) and scansorial taxa.” Why did they not break this category down into fully arboreal and scansorial categories? While I recognize that “scansorial” can be a vague category in itself, it seems that there would be considerable functional and morphological differences between the two groups. Even if the authors don’t choose to do this, could they speculate on how separating the arborealist category into these two groups would change results?

The organization of hypotheses and expectations is well done, however, each of these hypotheses seem to result in the same expected outcome: that gliders are between arborealists and flyers and that there is a functional throughline from arborealists -> gliders -> flyers. It feels a bit like all of your proposed hypotheses are shoehorned into this expectation and there are no real alternatives presented. I would suggest adding a separate model which is naive to your pre-determined optima. The ‘PhylogeneticEM’ package in R should be able to identify its own shifts in multivariate trait optima across a phylogenetic tree. This would be an opportunity for the data to show alternative optima.

Lines 230-233: Why was 69.8 MYA chosen for the placement of the fossil bat polytomy - can you justify this? Is it possible to mark this on the phylogeny in figure 1?

Validity of the findings

Other important parameters exist for OU model output and should be reported. The authors report theta values for optima, but alpha parameters should also be reported (which shows the strength of pull towards a particular optima), and the phylogenetic half-life (which shows the rate of adaptation). I would strongly recommend including these parameters in your results, as low alphas would indicate that a model with tall peaks on the adaptive landscape is not warranted. As reported, the results are mainly showing that the group means are different.

Additional comments

Thank you for the opportunity to review this manuscript exploring hypothetical functional transitions hypothesized to lead to the origin of powered flight in bats. The authors showed overall support for their main hypothesis, that gliders represent an intermediate form between arborealists and flyers (bats) based on linear measurements and multi-peak evolutionary models. They also showed support for previous findings that glider hindlimbs are generally longer but narrower than that of arborealists or flyers.

The authors had two main objectives: 1) to determine if glider forelimb traits have a separate, and intermediate, adaptive trait optima from arborealists and flyers; and 2) to identify the trait optima associated with hindlimb measurements for bats and gliding, arboreal, and ground-dwelling mammals. To investigate these objectives, they primarily used evolutionary models in several combinations of grouped ecomorphotypes to test the distinctiveness of limb trait optima, both with combined fore/hindlimb traits and at the individual measurement level. Overall, results supported the hypothesis that the forelimb traits of gliders are found at an intermediate adaptive peak between flyers and arborealists, and that the OU model with grouped arborealists and gliders was best supported. For hindlimb traits, distinct optima were found for each of the four locomotor groups, which highlights differing trends in hindlimb morphology.

Overall, I appreciate this contribution and find the approach an inventive way to ask questions about functional transitions with a lack of fossil information. I do, however, strongly recommend considering my general comments about the mammalian sample used, alternative hypotheses, the fossil bat polytomy placement, and reporting other OU model parameters. All code and relevant files should also be available for reproducibility purposes in addition to the raw measurement files. I have a few specific comments for the manuscript, figures, and tables below:

Notes on the manuscript with line numbers:
- Line 39: After this first sentence of the paragraph, could you elaborate on the bat fossil record a little bit? What does “uninformative” mean? That bat fossils are few and far between? That they cannot be attributed to particular lineages phylogenetically? That the fossil preservation is poor? I think that more discussion on the status of the bat fossil record would help to justify the approach you take in the paper.
- Line 45: What are the neighboring clades, specifically? Providing this context may help highlight the novelty of the bat ecomorphotype.
- Paragraph on lines 48-57: I think just after this paragraph you should introduce why hindlimbs are important to your analysis. How are hindlimbs incorporated into flight dynamics of bats vs gliding dynamics of other gliders? And with this, how does the patagium or wing membrane connect to the tail? Is there variation there that is influential to hindlimb morphology? This is just a suggestion - you discuss this in some ways in other parts of the paper, but I think highlighting the hindlimb up front might be good.
- Lines 124-127: Maybe tell us more about these fossil bats? Where/when did they come from and how might they be similar/different to modern bats? What is the existing knowledge we have?
- Line 135: “adding more phylogenetic history information” Do you actually have these bats in your phylogeny, and if so, how did you decide where they go? I think you address this later, but it would be good to talk about this earlier.
- Line 360: the name “Papadimitrious" is misspelled.
- Lines 663-664: Tadarida brasiliensis should be italicized.
- Line 708: “flight”? Is misspelled.

Notes on figures/tables/supplementary files:
Figure 1: The caption follows an “a) b) c)” format that is somewhat misleading - there are no “a” “b” “c” labels on the figure. I would suggest rewording the caption or updating the figure.
Figure 2: In a) the image of Rattus norvegicus is on top of a data point and cuts it off - this should be fixed. The genus and species names in the caption should also be italicized.
Figures 3 and 4: The green hues (for ground & arb., arborealists, and arb. & gliders) are a bit difficult to tell apart.
Table S7: Some rows are shaded and I’m not sure if that is intentional or not, or what that means.

---

## Round 0.2 · Minor Revisions

This manuscript has now been reviewed by both previous reviewers, who notably converge on the same finding: this revised manuscript is in great shape, and requires only minimal edits before being ready for publication. It will not need to be sent out for peer review again, but the authors should heed the reviewers' minor suggestions, including those I have spotlighted immediately here:

l. 98, Double quotes are preferably used for text quotations rather than single words, where it has the effect of making the individual word sound euphemistic. If reduced hindlimbs are actually what previous work articulated, citing it without quotes is completely fair and fine — instead, quotes in this case make the reader think it’s not actually true. Advise just stripping the double quotes away; the content is intact and the meaning is less prone to speculation.

l. 143, see above, l. 98. This issue is more overwrought with so-called transitional forms. As the case with early tetrapods and early whales, why not substitute with early or earliest bats? I can see an argument _for_ using euphemism with transitional forms — yes, it’s post hoc and typologic, but the point is that bats come out of no where, like ichthyosaurs and many other groups, so it conveys the point.

l. 261, take it easy on the fossils: they were once organisms and lineages in their own right. Traits, not taxa, are derived or plesiomorphic, right? As with l. 143, the point is taken; how about “fossil taxa show fairly derived traits (e.g., xx)”?

l. 263, delete pretty. Nothing lost in content and avoids a colloquialism.

ll. 504, 507, 523, 524 again, should at least say “fossil taxa” instead of merely “fossils”; but “fossil flyers” seems straightforward and clear. (Crown and stem terminology are incredibly useful in this way!).

l. 542, 579, clearly pulling is meant in the directional movement sense and does not need double quotes.

l. 583, please provide a reference for the sentence on phylogenetic half-lives. (Hansen 1997 or anything more recent is fine).

l . 616, see comment on 98.

·

Basic reporting

No comment

Experimental design

No comment

Validity of the findings

No comment

Additional comments

The authors have sufficiently addressed the comments from both reviewers. I have a few comments, mostly related to changes made in the previous round of revisions, but otherwise this manuscript is acceptable for publication. I look forward to seeing the published article!

Minor comments:
Line 43: “albeit” is misspelled

Line 145: change to something like “… only non-stem fossil bat in our analysis” since there are many known fossil taxa within the bat crown group.

Lines 504-506: you should probably point out that the crown families Tachypteron has been suggested to belong to (Emballonuridae or Miniopteridae) were not included in this analysis. Their inclusion might expand the range of living bats to encompass Tachypteron (but please note that I am not suggesting additional taxa need to be added to these analyses!)

Line 558: Galeopterus should be italicized

Line 563: Add a reference to Amador et al. 2019 here

Figure S1 caption: Remove “oldest” in reference to Onychonycteris as it is not the oldest fossil bat; this caption should also probably mention Tachypteron now that it has been highlighted in the figure

Reviewer 2 ·

Basic reporting

No comment

Experimental design

I think my questions about the extant sample and locomotor groups were answered well in the author response. They detailed how the ground-dwelling group is not variable in their trait optima, that allometric effects were removed, and that the scansorial and arboreal taxa are similar enough to be grouped together. I am glad that the authors included the PhylogeneticEM analyses, as I think this helped strengthen their results.

Validity of the findings

The suggestion to add the alpha and phylogenetic half-life parameters was addressed well in the manuscript.

Additional comments

I thought the authors did a great job with incorporating the reviews, and I particularly like the new figure showing evolutionary shifts in forelimb and hindlimb morphology. All of my initial comments on this manuscript were addressed in detail in the review response and manuscript. I have no remaining concerns. Thank you for providing the R scripts. Here are some specific edits (PDF):
Line 558: Galeopterus should be italicized.
Line 877: Should the phrase "The forelimb measurement consisted..." be "The forelimb measurements consisted..."?

---

## Round 0.3 · accepted · Accept

Congratulations! This final revised manuscript has cleared all the hurdles and it's ready to go: both of the reviewers' comments were accepted, along with the Editor's (mine). This manuscript is ready for publication! Thank you for your contribution to understanding bat origins!